# *OsPUB9* Gene Edited by CRISPR/Cas9 Enhanced Resistance to Bacterial Leaf Blight in Rice (*Oryza sativa* L.)

**DOI:** 10.3390/ijms25137145

**Published:** 2024-06-28

**Authors:** Me-Sun Kim, Van Trang Le, Yu Jin Jung, Kwon-Kyoo Kang, Yong-Gu Cho

**Affiliations:** 1Department of Crop Science, College of Agriculture and Life & Environment Sciences, Chungbuk National University, Cheongju 28644, Republic of Korea; kimms0121@chungbuk.ac.kr (M.-S.K.); vantrang.mekong@gmail.com (V.T.L.); 2Division of Horticultural Biotechnology, Hankyong National University, Anseong 17579, Republic of Korea; yuyu1216@hknu.ac.kr

**Keywords:** ubiquitination, E3 ligase, *OsPUB*, CRISPR/Cas9, bacterial leaf blight

## Abstract

Ubiquitination plays a crucial role in regulating signal pathways during the post-translation stage of protein synthesis in response to various environmental stresses. E3 ubiquitin ligase has been discovered to ultimately control various intracellular activities by imparting specificity to proteins to be degraded. This study was conducted to confirm biological and genetic functions of the U-box type E3 ubiquitin ligase (PUB) gene against biotic stress in rice (*Oryza sativa* L.). *OsPUB9* gene-specific sgRNA were designed and transformants were developed through *Agrobacterium*-mediated transformation. Deep sequencing using callus was performed to confirm the mutation type of T_0_ plants, and a total of three steps were performed to select null individuals without T-DNA insertion. In the case of the *OsPUB9* gene-edited line, a one bp insertion was generated by gene editing, and it was confirmed that early stop codon and multiple open reading frame (ORF) sites were created by inserting thymine. It is presumed that ubiquitination function also changed according to the change in protein structure of U-box E3 ubiquitin ligase. The *OsPUB9* gene-edited null lines were inoculated with bacterial leaf blight, and finally confirmed to have a resistance phenotype similar to Jinbaek, a bacterial blight-resistant cultivar. Therefore, it is assumed that the amino acid sequence derived from the *OsPUB9* gene is greatly changed, resulting in a loss of the original protein functions related to biological mechanisms. Comprehensively, it was confirmed that resistance to bacterial leaf blight stress was enhanced when a mutation occurred at a specific site of the *OsPUB9* gene.

## 1. Introduction

U-box E3 proteins play a regulatory role in protein degradation through the proteolytic mechanism of multi-protein E3 ubiquitin ligase in response to cellular signals during plant development and growth, hormone response, and stress responses [1,2]. *AtPUB*-ARM E3 ubiquitin ligase interacts with S-domain receptor kinases involved in ABA signaling [3], and *AtPUB22* interacts with MITOGEN-ACTIVATED PROTEIN KINASE3 (*MPK3*) to control the immune response [4]. The existence of various types of U-box E3 gene families indicates that they are involved in diverse biological functions [5]. Many studies have revealed that plant U-box E3 genes play important roles in the regulation of plant development. The pub4 mutants in Arabidopsis showed higher levels of cell proliferation and division in root and shoot apical meristems. The root apical meristem of pub4 exhibited a decrease in inhibition of root cell proliferation and stem cell maintenance by CLAVATA3 (*CLV3*). Additionally, plants utilize the ubiquitin proteasome system as a rapid mechanism to activate pathogen-related proteins and induce plant immunity regulation via pattern recognition receptors (PRRs) to respond to a pathogen attack. *OsBBI1*, which is involved in rice blast resistance through its E3 ubiquitin ligase activity, is induced by *magnaporthe oryzae* and hormones such as salicylic acid, which regulate resistance to blast disease mainly by regulating cell wall thickness [6,7]. *OsJAZ8* with a truncated Jas domain exhibited resistance to rice white leaf bacterial blight by degradation of *OsJAZ8* through the ubiquitin-proteasome pathway in exogenous jasmonic acid treatment. *AvrPizt*, a rice blast-related effector protein, induces plant immune responses by suppressing the activity of *APIP6*, a RING type E3 ubiquitin ligase, and *WRKY11*, a transcription factor that promotes disease resistance regulation, and interacts with *EIRP1*, a RING type E3 ubiquitin ligase, to induce plant immune response in grapes. It has been shown to improve disease resistance [8,9,10]. Past studies have revealed that many E3 ubiquitin ligases are diversely involved in plant immunity-related substrates, the characteristics of the ubiquitination they catalyze, and the basic mechanisms involved in plant immunity.

Non-homologous end joining (NHEJ) is a primary double-strand breaks (DSBs) repair pathway in somatic plant cells and is therefore relatively easy to harness for genome editing [11]. This repair pathway is error-prone and often introduces a small insertion or deletion (InDel) at the break site, which can shift the open reading frame of a coding sequence or create a premature stop codon, resulting in a gene knockout. If DSB occurs within a protein-coding region of the gene, InDels result and can produce frameshift mutations, changing the reading frame and possibly producing a non-functional protein. In some cases, NHEJ can also introduce stop codons, leading to premature termination of protein synthesis. Gene knockouts or disruption of particular gene functions can be produced by using NHEJ of the CRISPR/Cas9 system. By introducing targeted DSBs and relying on cell repair machinery, researchers can induce gene modifications and study consequences of gene inactivation. The CRISPR/Cas9 system has been successfully applied for gene knockout in the study of gene function, as well as for engineering various traits including disease resistance [12], altered development [13,14], male sterility [15], higher yield [16], and improved nutrition [17]. Moreover, multiplex CRISPR/Cas9 systems have enabled simultaneous knockout of multiple genes [18]. Recent research using this strategy has highlighted the potential role of AP2/ERF domain-containing RAV (related to ABI3/VP1) TF family members in abiotic stress adaptation [19]. In cases where a complete knockout is desired, gene deletion can be achieved by utilizing a multiplex CRISPR/Cas9 system that targets both 5′ and 3′ ends of gene. Furthermore, multiplex CRISPR/Cas9 systems offer capability to generate extensive mutations such as chromosomal deletions and translocations. Notably, chromosomal deletions spanning (115–245 kb) have been successfully created to eliminate gene clusters in rice [20].

Bacterial leaf blight poses a significant threat to rice production in both tropical and temperate regions. This destructive disease is caused by *Xanthomonas oryzae* pv. *oryzae* (*Xoo*), which leads to necrosis in rice leaf blade, ultimately reducing the photosynthetic rate and impacting grain yield [21,22]. Despite extensive breeding efforts aimed at combating this disease, its widespread occurrence persists in major rice-growing areas. Bacterial leaf blight can inflict damage at any stage of the rice-growing cycle. The disease is notorious for causing significant yield losses, typically ranging between 10% and 30%. However, in some cases, the losses can be as high as 80%, depending on various factors such as location, season, weather conditions, crop growth stage, and the specific rice cultivar [23,24,25,26]. To date, researchers have identified at least 45 resistance genes from both wild and cultivated rice accessions, among these genes, *Xa4*, *xa5*, *Xa7*, *xa13*, and *Xa21* have been commonly deployed in rice farms across Asia [27,28,29]. However, the large-scale and long-term cultivation of certain resistant cultivars has led to significant shifts in the race frequency of *Xoo* and the breakdown of single major resistance genes, such as *Xa4* [29,30,31]. Since 2003, the widespread occurrence of *Xoo* race K2 in Korea has been observed. This particular strain or pathotype of *Xoo* possesses distinct characteristics and virulence factors [24]. These challenges highlight the need for continued research and development of new strategies to combat bacterial leaf blight. It is essential to identify and deploy new sources of resistance, explore combinations of multiple resistance genes, and implement integrated disease management approaches to mitigate the disease’s impact and safeguard rice production in the face of evolving pathogen populations. In the Kitaake context, however*, OsSWEET14* single-deletion or promoter mutants are weakly resistant or even vulnerable to African *Xoo* strains. CRISPR/Cas9 was used in another research to impair the function of *OsSWEET14* by altering its equivalent coding sequence in rice Zhonghua 11. It was found that plant height was increased without decreasing yield when *OsSWEET14* was disrupted [32].

In a recent study, researchers used the CRISPR/Cas9 system to change sequences in all three SWEET gene promoters (*SWEET11*, *SWEET13*, and *SWEET14*). Sequence analysis of TALe (transcription-activator-like effectors) genes in 63 *Xoo* strains revealed numerous TALe variations for *SWEET13* alleles, which aided gene editing. *SWEET14*, which is similarly targeted by two TALes from an African *Xoo* lineage, has been mutated. Kitaake, IR64, and Ciherang-Sub1 rice varieties were all given five promoter mutations simultaneously. Paddy trials revealed that these rice lines with genome-edited SWEET promoters have broad-spectrum and robust resistance [33]. In addition, the CRISPR/Cas9 technology was used to delete rice gene *Os8N3* to improve *Xoo* resistance. These mutations were transmitted to the next generations and genotyped. Rice plants in T_0_, T_1_, T_2_, and T_3_ generations were edited, and the result showed that the homozygous mutants had significantly enhanced resistance to *Xoo*. The T_1_ generation demonstrated stable CRISPR/Cas9-mediated *Os8N3* gene editing transmission without the transferred DNA [34]. In this study, the main objective was focused on evaluating phenotypes of the T_2_ *OsPUB9* gene-edited null lines under bacterial leaf blight stress conditions in rice. Our study shows that two *OsPUB9* gene-edited null lines, GE.PUB9-3-5, and GE.PUB9-3-6, exhibited a surprisingly strong resistance phenotype through phenotypic analysis of infection sites and analysis of histochemical staining. The lesion lengths observed in both null lines were very similar to those in Jinbaek, demonstrating the effect of resistance to *Xoo* infection. These findings suggest that two *OsPUB9* gene-edited null lines selected in this study will provide potential usefulness in solving important problems caused by bacterial leaf blight stress.

## 2. Results

### 2.1. Generation and Characterization of Null Mutants through CRISPR/Cas9-Targeted Knockout of the Ospub9 Gene

#### 2.1.1. Generation of Knockout Mutant Lines of the *OsPUB9* Gene

The *OsPUB9* gene related to biotic stress among U-box proteins and located in chromosome 2 were reviewed and target locations were selected for CRISPR/Cas9 application. The *OsPUB9* genes consisted of four exons and three introns. Target sgRNA to edit the *OsPUB9* gene using the CRISPR/Cas9 system was selected from the fourth exon (120 bp away from the third intron flanking region) (Figure 1A). As a result, the final sgRNA selection targeted the base sequence of 20 nucleotides including the NGG sequence in the 3′ region of exon 4 consisting of 260 bp. Among them, sequences with a GC content of 30 to 70%, an out-of-frame score of 50 or more, and a mismatch of 1-0-0 are first selected, and then the sequence with the highest out-of-frame score is finally selected (Table 1).

In constructing of a plant expression Ti-plasmid vector into which CRISPR/Cas9 was introduced, a pPZP-Cas9 vector containing Cas9 and bar selection markers were constructed under the control of the CaMV35S promoter. Next, a final vector was constructed to express a gene under the *OsU3* promoter by cloning sgRNA into the *Aar* I site. Colonies were selected after transformation into *E. coli*, plasmid DNA was extracted, and sgRNA introduction was confirmed. As a result, PCR bands at 570 bp were shown, confirming sgRNA introduction. The vectors for which sgRNA introduction was primarily confirmed by sequencing analysis were commissioned to the Bioneer sequencing service (Bioneer Corporation, Daejeon, Republic of Korea) and were correctly constructed as target sgRNAs and used for plant transformation (Figure 1B). The CRISPR/Cas9-OsPUB9 vector was inoculated into callus obtained from Dongjin (wild-type) seeds using an *Agrobacterium*-mediated method to generate a total of 9 T_0_ transformants for the *OsPUB9* gene. PCR analysis was performed by extracting gDNA from T_0_ generation in order to confirm successful transformation by confirming introduction of T-DNA. As a result, it was confirmed that all nine regenerated plants transplanted into soil were transformants into which the CRISPR/Cas9-OsPUB9 vector was stably introduced by an *Agrobacterium*-mediated method (Appendix A).

#### 2.1.2. Analysis of Mutation Types through Deep-Sequencing and Generation of *OsPUB9* Gene-Edited Null Lines

The presence or absence of gene editing and the type of mutation in nine individuals transformed with the CRISPR/Cas9-OsPUB9 vector were analyzed using NGS technology. Among nine transgenic rice plants, a total of five plants had mutations in *OsPUB9* gene regions, and the average efficiency of gene editing among transgenic rice plants was 55.6%. Among five mutants, one individual had a homozygosity genotype, and four individuals had a bi-allelic genotype (Figure 1C, Appendix A).

T_1_ seeds were harvested by cultivating individuals identified as gene editing plants. Null individuals from which T-DNA was removed were selected using bar gene. From the T_1_ generation, null individuals without T-DNA were selected through bar screening. Null individuals whose bar gene were found to be removed through treatment with herbicide basta 40 ppm and results of bar strip and bar gene PCR were selected. Finally, according to results of the genetic segregation ratio test through the chi-square (χ^2^) test, a single copy introduction was confirmed at 3:1 (Appendix A, Appendix A).

#### 2.1.3. Analysis of Putative Protein Expression Based on Gene Editing in Null Mutants

The amino acid sequence of *OsPUB9* gene-edited null lines was reconstructed using a translator tool (accessed on 24 January 2024, https://web.expasy.org/translate/) based on an edited nucleotide sequence (Appendix A). In the case of GE.PUB9-3-5, a one bp insertion occurred by gene editing, and it was confirmed that early stop codon and multiple ORF (open reading frame) sites were created due to the insertion of thymine (T) (Figure 2A).

Based on the previously analyzed amino acid sequence, in order to determine whether changes in bases in *OsPUB9* gene-edited null line causes any changes in *OsPUB9* protein structure, we examined the *OsPUB9* domain structure and key functional residues using the protein modeling program. In the GE.PUB9-3-5 line where early stop codon and many ORFs occurred, isoleucine was changed to serine in the 381st amino acid sequence of the *OsPUB9* gene, which plays an important role in *OsPUB9* gene domain structure. Accordingly, the subsequent peptide chains were connected with the ARM repeat domain in the wild-type *OsPUB9*, whereas in *OsPUB9* of GE.PUB9-3-5 they were not connected to the ARM repeat domain (Figure 2B).

Also, the *OsPUB9* gene-edited null line showed significant differences in the *OsPUB9* protein structure due to the effect of inserting one SNP in the *OsPUB9* motif domain and the changing amino acid (Figure 2C). Therefore, this suggests that the newly formed *OsPUB9* protein in the *OsPUB9* gene-edited null line may act differently from protein function related to biological mechanism.

### 2.2. Biotic Stress Resistance in OsPUB9 Gene-Edited Null Lines

#### 2.2.1. Evaluation of Resistance against *Xanthomonas. Oryzae v. oryzae*

Interesting results were obtained when T_2_ *OsPUB* gene-edited null lines were screened 14 days after bacterial blight inoculation using *Xoo* Korean race K2 (Appendix A). Dongjin (WT) showed clear sensitivity with a lesion length of 7.3 cm. On the other hand, Jinbaek (right) showed strong resistance with a significantly reduced lesion length of 0.5 cm. Most of T_2_ *OsPUB9* gene-edited null lines showed similar sensitivity to Dongjin (Figure 3).

However, two lines, GE.PUB9-3-5 and GE.PUB9-3-6, showed a marked phenotype of strong resistance. The lesion length observed in these lines was very similar to that of Jinbaek, indicating a resistance effect against *Xoo* infection (Appendix A).

#### 2.2.2. Analysis of Histochemical Staining

DAB staining can detect H_2_O_2_, one of the reactive oxygen species (ROS) that can be produced in plants by various stresses [35]. When leaves collected 24 h after treatment with bacterial leaf blight K2 strain were stained with DAB, browner H_2_O_2_-DAB reaction products were identified in wild-type tissues (Figure 4). These results show that at a certain point in time when the wild type is subjected to bacterial leaf blight stress, there is more H_2_O_2_ in the tissue compared to the *OsPUB9* gene-edited null lines. Therefore, GE.PUB9-3-5 and GE.PUB9-3-6 are induced by resistance to bacterial leaf blight stress.

### 2.3. Expression Analysis of R Genes Related to Biotic Stress Resistance

We aimed to identify the expression of R genes related to resistance response to bacterial blight stress. Extensive genetic studies on resistance to bacterial blight stress have been conducted over the last 20 years. More than 40 resistance (R) genes conferring host resistance to various strains of *Xoo* have been identified and 11 of them were cloned [36]. These R genes can be classified into four groups based on their encoding proteins, including receptor-like kinase (RLK) genes (*Xa21*, *Xa3/Xa26* and *Xa4*), sugar will eventually be exported transporter (SWEET) genes (*xa13*, *xa25* and *xa41*), executor genes (*Xa10*, *Xa23* and *Xa27*) and other types of genes (*Xa1* and *xa5*) [36]. We have tested eleven R genes and one PUB gene to check their expression with the GE.PUB9 gene-edited rice line. These genes used for gene expression comparison each consist of four RLK genes (*XB3*, *XB25*, *XB21*, and *XB24*), two SWEET genes (*xa14* and *xa41*), three executor genes (*Xa10*, *Xa24*, and *Xa27*), and three other genes (*Xa1*, *xa5*, and *PUB22*).

Genes belonging to group 1 that showed strong expression in GE.PUB9-3-5 have a leucine-rich repeat receptor-like kinase (*LRR-RLK*) structure and are known to maintain the stability of *Xa21* as *Xa21* binding protein (XB). In group 2, among the recessive R genes *xa13* and *xa41, xa13* was confirmed to be expressed in the gene-edited line. The three executioner genes are structures with various potential transmembrane domains and are transcriptionally activated by TAL effectors and play a role in triggering defense responses. It was represented that the *Xa27* gene showed strong expression even before treatment with the bacterial leaf blight K2 strain in GE.PUB9-3-5. Of the eleven R genes, nine R genes revealed high expression 12 h after inoculation of bacteria in the GE.PUB9-3-5 line, implying that those R genes are highly related to the expression of high resistance to bacterial blight induced with the knocking out of *OsPUB9* (Figure 5).

### 2.4. Investigation of Agronomic Traits

To observe the growth of T_3_ *OsPUB9* gene-edited null lines in a rice field, major agricultural traits such as plant height, culm length, panicle length, and number of tillers were investigated (Table 2, Figure 6). *OsPUB9* gene-edited null lines did not differ from the WT in terms of the four agricultural traits, suggesting that gene editing using the CRISPR/Cas9 technology did not significantly affect genes other than *OsPUB9*.

## 3. Discussion

Ubiquitination plays a pivotal role as a major form of post-translational modification in regulating plant growth, development, and the plant’s response to stresses [37]. The E3-ubiquitin ligase stands out as a primary determinant of the fate of target proteins, consequently influencing their physiological effects [38]. Among the various enzymes involved in ubiquitination, E3-ubiquitin ligases are notably abundant. It is predicted that the genomes of Arabidopsis, rice, and maize collectively harbor over 1100 genes encoding E3-ubiquitin ligases [39,40]. Numerous studies have highlighted the significant involvement of ubiquitination components, particularly monomeric E3-ubiquitin ligases, in the rice plant’s response to diverse stresses. The abundance of these ligases is thought to be associated with their specificity for target proteins [41]. The CRISPR/Cas9 system has been used as a powerful tool to investigate gene function and produce new mutants in crop productivity. Mutations resulting from double-strand break repair in plants appear mainly as deletions and insertions. These mutations exhibit stable inheritance in subsequent generations, including both T_2_ and T_3_, following the classical Mendelian model. In cases where both copies of the target gene are mutated, such as in homozygous or bi-allelic editing plants, the genotype remains stable and resistant to further editing by CRISPR/Cas9. Therefore, additional breeding steps such as selfing and/or crossing are often needed to combine null mutations or make them homozygous [42,43].

In this study, our goal was to identify the function of the *OsPUB* gene related to stress mechanism by knocking out of *OsPUB* genes at specific sites to induce mutations by the CRISPR/Cas9 system. Our hypothesis was that applying the NHEJ method for knocking out the *OsPUB* gene might result in targeted bi-allelic frameshift mutations, ultimately leading to enhanced resistance to biotic stresses in rice. The *OsPUB9* genes were knocked out through the CRISPR/Cas9 system and showed editing efficiency (55.6%) consistent by the NHEJ pathway. In the case of the *OsPUB9* gene-edited null line, a one bp insertion occurred by gene editing, and it was confirmed that early stop codon and multiple ORF sites were created due to the insertion of thymine (Figure 1C). During the analysis of the domain structure and protein modeling, in GE.PUB9-3-5 where many ORFs occurred, it is assumed that the ubiquitination function also changed according to the domain region and protein structure change unlike *OsPUB9* in wild type (Figure 2). Therefore, it was estimated that the *OsPUB9* gene and deduced amino acid sequences were greatly changed and the protein function related to the biological mechanism may be changed. According to previous studies on changes in protein function due to single-nucleotide mutations, *HDAC10* accumulates in *osprr73* mutants (1 bp insertion) and transcriptionally represses *OsHKT2-1*, thereby conferring salinity tolerance and reducing Na+ accumulation in cells [44]. In addition, there was a study that reported that the salinity tolerance of rice was improved by changing the function of the protein through nucleotide insertion using a Cas9-OsRR22-gRNA expression vector targeting the *OsRR22* gene of rice [45].

In inoculation of the bacterial leaf blight K2 strain, GE.PUB9-3-5 and GE.PUB9-3-6 showed shorter lesion lengths than Dongjin and this lesion length observed in two *OsPUB9* gene-edited null lines was very similar to that of the resistant variety, Jinbaek (T) (Figure 3). It was identified that more brownish H_2_O_2_-DAB reaction products were generated in wild-type tissues after treatment of the *Xoo* K2 strain (Figure 4). These results show that GE.PUB9-3-5 and GE.PUB9-3-6 induced high-resistance mechanism by exposure to bacterial leaf blight stress and strong resistance to oxidative stress can be acquired. *Avr3a*, an effector of the potato blight pathogen *P. infestans*, was shown to interact with the *CMPG1* gene and prevent host cell death during the biotrophic phase of *N. bethamiana* [46]. Similarly, the *XopP* effector from rice bacterial blight *Xoo* was shown to interact directly with the *PUB44* U-box domain and inhibit autoubiquitination in vitro [47]. Transient overexpression of *XopP* led to the accumulation of *OsPUB44* and was shown to suppress peptidoglycan (PGN) and chitin-induced immunity and *Xoo* resistance. To identify the expression of R genes related to a resistance response to bacterial blight stress, we have examined the R genes related to a resistance to bacterial blight (Figure 5). We have tested twelve R genes to check their expression with the *OsPUB9* gene-edited rice line. Of all the R genes, seven R genes revealed high expression 12 h after inoculation of bacteria in the GE.PUB9-3-5 line, implying that those R genes are highly related to the expression of high resistance to bacterial blight induced with the knocking out of *OsPUB9*. Comprehensively, knock-out by CRISPR/Cas9 technology in this study caused a loss in the structural and biological function of the *OsPUB9* gene. According to previously reported research, in the case of dicotyledonous plants, the AtU6 promoter derived from *Arabidopsis* is attached to the front of sgRNA to induce regulation, and it is reported that gene editing efficiency can be increased with the CRISPR/Cas9 enzyme [48,49]. A vector using sgRNA and the *OsU3* promoter was constructed to create rice mutants, and it is believed that an effective mutation frequency was found.

In Figure 5, we examined the expression levels of eleven R genes and found that the expression level of R genes was high in the *OsPUB9* gene-edited null line. In addition, similar results were obtained in the DAB experiment regarding the response to oxidative stress over time after *Xoo* treatment. This suggests that the *OsPUB9* gene may play a significant role in the biotic stress response. Typically, the *OsPUB9* gene play various roles in plant cells due to the presence of the U-box/ARM domain, and actively interacts with proteins [50]. *SPL11* containing a U-box domain and ARM domain is involved in ubiquitination and protein interactions and relies on the typical U-box domain for ubiquitination to regulate cell death and trigger [51]. *EL5*, a RING type E3 ubiquitin ligase, was found to bind and interact with *CsUBC5b*. EL5C153 and EL5W165A, which lack coding activity due to the absence of the RING domain, were shown to result in a short root or rootless phenotype, suggesting the importance of *EL5* in root primordial cells [52,53]. Interestingly, considering that the *OsPUB9* knock-out mutant that does not constitute complete domains obtained through this study, showed resistance to bacterial blight, it is thought to be closely related to the R gene. Finally, to gain a deeper understanding of the PUB function, gene structural analysis is believed to play an important role in deciphering the complex interactions between domains present in PUB. The need for structural information is most urgent for PUB, which consists of UND, U-box, and ARM repeats because, unlike highly conserved E3 ubiquitin ligases such as *Prp19*, Ufd2, or *CHIP*, it is unique to higher plants [54].

Extensive reports have consistently demonstrated that transgenic plants can acquire a stress tolerance phenotype through modulation of stress-responsive E3-ubiquitin ligases, and this can be achieved by either overexpressing or suppressing E3-ubiquitin ligases. However, research of ubiquitination pathway still faces various challenges that require further investigation to advance this field. Notably, study of membrane-associated E3-ubiquitin ligases have a distinct set of obstacles. Traditional experiments like yeast two-hybrid (Y2H) or bimolecular fluorescence complementation (BiFC) are not readily applicable due to the unique characteristics of membrane proteins. Y2H relies on nuclear translocation, which is not observed in membrane proteins, while BiFC can disrupt protein structure and hinder binding due to the introduced fluorescence fusion tag. Nevertheless, alternative and complementary methods such as the split ubiquitin system [55], proximity labeling [56], and ratiometric BiFC (rBiFC) [57] have emerged to address these challenges and have proven extremely valuable in elucidating the complex interactions of E3-ubiquitin ligases. If these methods overcome hurdles in studying E3-ubiquitin ligases, it will be possible to perform comprehensive functional characterization of *OsPUB* genes and their interacting genes, leading to a deeper understanding of their molecular and physiological importance in rice. In conclusion, results obtained in this study provide valuable insights into the expression dynamics of the *OsPUB9* gene under biotic stress, and also provide a solid foundation for study of the underlying molecular mechanisms and mechanism information for gene regulation involved in complex interactions between stress responses.

## 4. Materials and Methods

### 4.1. Selection of Target Sites and sgRNA of the OsPUB9 Gene

Genomic sequences and coding region sequences were searched in Gramene (accessed on 24 January 2024, http://www.gramene.org), Rice Genome Annotation Project (accessed on 24 January 2024, http://rice.uga.edu/), and NCBI (accessed on 24 January 2024, http://www.ncbi.nlm.nih.gov/gene). Target sites and sgRNA of *OsPUB9* (Os02g0732200) adjacent to a protospacer-adjacent motif (PAM) were designed using the CRISPR RGEN tool (accessed on 24 January 2024, http://www.rgenome.net/) developed by HanYang University [58]. Targeting sites were selected from the CDS region of the *OsPUB9* gene, and sgRNAs were obtained by using the CRISPR RGEN tool program. First, SpCas9′s 5′-NGG-3′PAM type derived from Streptococcus pyogenes in the Cas9-designer search tool was selected to apply to the Cas9 enzymes. Next, sgRNA was designed by entering the exon sequence of the *OsPUB9* gene in the target sequence input box after selecting Oryza sativa as the target genome. The sgRNAs with a GC content of 35–65%, out of the frame score from 73.9 to 89.4, and no mismatches in the genome were selected, which can reduce the off-target probability. RGENs of the selected *OsPUB9* gene were used as a vector for rice transformation to introduce into the wild type (WT; *Oryza sativa* L. cv. Dongjin).

### 4.2. Synthesis of sgRNA and Construction of CRISPR/Cas9 Vector

The selected sgRNA was synthesized by designing a complementary two-stranded oligonucleotide by attaching adapters of the restriction enzyme Aar I to the front and back of 20 base pairs excluding the PAM region for vector cloning [sgRNA up: 5′-GGCAG (sgRNA 20 bp Fw)-3′, sgRNA down: 5′-AAAC (sgRNA 20 bp Rv) C-3′]. The transformation vector was constructed to introduce the selected *OsPUB9* RGEN into the WT. The pPZP-3′PINII-Bar vector was used as the parent vector for plant transformation. Cas9, under the *CaMV 35S* promoter, and sgRNA, under the *OsU3* promoter, were ligated to this vector. Bar, under the control of the *CaMV 35S* promoter, was used as the selection marker. To verify the successful introduction of sgRNA, a PCR analysis was conducted using 1 μL of the culture medium. The PCR product was then subjected to digestion using the Aar I enzyme, and the resulting cleavage was confirmed by electrophoresis. Bands showing at the 670 bp region were selected, and the binary constructs were then introduced into the *Agrobacterium tumefaciens* strain EHA105 via electroporation [59].

### 4.3. Transformation to Rice Callus via Agrobacterium-Mediated Method

*Agrobacterium*-mediated transformation of embryogenic callus was performed according to the description previously reported by Lee et al. [59] with some modifications. For *Agrobacterium*-mediated rice transformation, good seeds of WT were used and surface sterilization by sodium hypochlorite (NaOCl) was performed. The callus was grown for 3 weeks in 2N6 callus-inducing media at 30 °C in 16 h light and 8 h dark conditions. After three weeks, the embryogenic calli were harvested, and inoculated with 0.1 OD of transformed *Agrobacterium* suspension. After co-cultivation, the callus was washed gently with sterilized water added with carbenicillin and then plated in 2N6-CP-selecting medium and were sub-cultured to fresh media every two weeks. After co-cultivation, the callus was washed gently with sterilized water added with carbenicillin and then plated in 2N6-CP-selecting medium and were sub-cultured to fresh media every two weeks. During this period, selection of regeneration calli in MSR media was carefully observed and handled. After forming the plantlet formed in MS media, regenerated rice seedlings were acclimatized for two days in greenhouse conditions before being transferred into soil and transferred to pots in a greenhouse maintained at 30 °C during the day and at 23 °C at night.

### 4.4. Analysis of Mutation Types Using Deep-Sequencing and Selection of Null Mutants

For genotype analysis of the target site and the selection of mutants, genomic DNA was extracted from the leaves of T_0_ plants where T-DNA introduction was confirmed, and deep sequencing was performed. NGS analysis was performed to amplify the genomic region containing the CRISPR/Cas9 target sites using specific primers adjacent to the designed target site [60]. PCR amplicons were sequenced by forming paired end reads using MiniSeq (Illumina, San Diego, CA, USA). The NGS data obtained were analyzed using the Cas-Analyzer (accessed on 24 January 2024, http://www.rgenome.net/cas-analyzer/) [61]. Insertion and deletion mutations were considered as mutations induced by Cas9. *OsPUB9* gene-edited plants with specific mutations in the target site were selected and T_1_ generation was produced for null segregation selection [62]. To select T_0_ null lines without T-DNA, 14-day-old seedlings with more than six leaves were treated with 40 ppm of basta (glufosinate ammonium) about 3 cm from the tip of the leaf. The basta solution-treated plants were observed after seven days to distinguish between resistance and sensitivity and were used to reconfirm whether they were resistant or sensitive using the bar strip test (AgraStrip^®^ LL Bulk Grain Strip Test, Romer Labs, Tulln, Austria). T-DNA-free plants were selected on the basis of the death of the leaves after basta treatment and the absence of bands in the bar strip experiments. Finally, PCR analysis was performed on null lines clearly lacking the T-DNA using a bar primer, and individuals with no band in 1% agarose gel electrophoresis were selected as the final null T_2_ lines.

### 4.5. Pathogenicity Test of Leaf Bacterial Leaf Blight in Null Mutants

Six-week-old T_2_ *OsPUB9* gene-edited null lines, wild-type Dongjin (susceptible variety), and Jinbaek (resistant variety) were inoculated with *X. oryzae* pv. *oryzae* strain K2 resuspended in water at a density of 10^5^ CFU/mL using the leaf-clipping method [63]. Bacterial inoculum was prepared from a 48 h-old culture of the *Xanthomonas oryzae* pv. *oryzae* strain K2 grown on a plated peptone sucrose agar medium. Each plant was cut from the tip of the leaf to about 2 to 5 cm and inoculated with the K2 strain. Plants inoculated with the *Xoo* K2 strain were grown under appropriate conditions such as 28–32 °C temperature, 13 h lightness/11 h darkness, and 90% relative humidity and the progression of the lesions was observed. The mean lesions of 3 plants (2 leaves per plant) with five replications were recorded at 14 days after inoculation. Two weeks after inoculation, the length of the lesion was measured and statistically processed, and the phenotype of the lesion site was observed and scored. The lesion length was measured from the inoculated site to the edge of the midvein of the thoroughly bacterial blighted leaf. Scoring was carried out by measuring the lesion length in cm. Resistance data against the bacterial blight were taken at 14 days after inoculation, following the Standard Evaluation System (SES) [64].

### 4.6. Analysis of Putative Protein Expression in Null Mutants

To analyze the protein expression due to the frameshift of the *OsPUB9* gene-edited null lines where InDels occurred, the nucleotide sequence was reconstructed based on the results of the NGS analysis. Amino acid sequence information was obtained through the translator tool (https://web.expasy.org/translate/, accessed on 24 January 2024) provided by Expasy using the reconstituted nucleotide sequence. To confirm the structure of the protein according to the amino acid sequence, a modeling program of the SBI (https://swissmodel.expasy.org/interactive, accessed on 24 January 2024) was used for the amino acid sequence obtained above.

### 4.7. Analysis of DAB Staining

3,3′-diaminobenzidine tetrahydrochloride (DAB) staining was performed to determine the generation of reactive oxygen species (ROS), a phenomenon observed when cell death is induced [35]. The *Xoo* K2 strain was cultured and resuspended in water at a density of 10^5^ CFU/mL and inoculated into plants using the leaf-clipping method. Each seedling was inoculated for three replications, starting with the flag leaves first and continuing up to the third leaf. On the 14th day after treatment with the *Xoo* K2 strain, the middle 2 cm part of the third leaf of each plant was collected, immersed in a 0.1% DAB (pH 3.8) solution, and treated for 1 h under dark conditions at 25 °C. After DAB staining, the sample was bleached four times with 94% ethanol heated to 70 °C, and brown H_2_O_2_-DAB reaction tissue deposited on the tissue was observed using a dissecting microscope.

### 4.8. Expression Analysis of R Genes Related to Biotic Stress Resistance

The upper part of the leaf was collected at 0, 6, and 12 h after K2 strain inoculation in wild-type and gene-edited null lines, respectively. The leaf samples were placed in a 2.0 mL tube with liquid nitrogen, and it was ground with a blue pencil. The total RNA was extracted using the TRIzol^®^ reagent (Invitrogen, CA, USA) according to the manufacturer’s instructions. The quality and concentration of the extracted RNA were quantified by the NanoDrop 2000 spectrophotometer (Thermo Fisher, Waltham, MA, USA) and the RNA integrity was determined by agarose gel electrophoresis. One mL of TRIzol^®^ reagent was added to the tube containing the sample after being grinding, homogenized for five minutes, centrifuged at 4 °C at 12,000× *g* for 10 min, and the supernatant was transferred to a new 1.5 mL tube. Subsequently, 200 μL of chloroform was added to the homogenized sample, thoroughly mixed with a vortex mixer for 15 s, and then reacted at room temperature for two minutes. After centrifugation at 12,000× *g* at 4 °C for 15 min, the supernatant was transferred to a new tube and 400 μL of isopropanol was added, inverted, and thoroughly mixed. The solution in the micro-tube was removed by centrifugation and further centrifuged at 10,000× *g* for one minute at room temperature to remove the remaining ethanol. Then, 50 μL of nuclease-free water was added to the micro-tube, left to react for one minute, and then centrifuged at 10,000× *g* for one minute at room temperature to collect the RNA extracted from the tube. The extracted RNA was stored at −70 °C.

The specific sequences of primer pairs were used in a semi-quantitative reverse transcription PCR (RT-PCR). The primers used are detailed in Appendix A. The RT-PCR was performed at 94 °C for 3 min for 1 cycle and 94 °C for 30 s, 5n °C (adjusted according to each primer) for 30 s, and 72 °C for 40 s for 35 cycles. The amplified cDNA was separated by electrophoresis using 1.4% agarose gel. *Actin* primers were used as control to normalize the results of the RT-PCR.

### 4.9. Investigation of Agronomic Traits

Three independent T_3_ *OsPUB9* gene-edited null lines and wild-type Dongjin were evaluated for morphological characteristics under the field condition. The fertilizer N-P_2_O_5_-K_2_O was applied at a rate of 90-45-47 kg/ha. Management of cultivation was performed following the rice cultivation standards adapted to the experimental farm of Chungbuk National University. Morphological characteristics including plant height, culm length, panicle length, and number of tillers were determined at the mature stage. All the measurements were replicated at least three times.

## 5. Conclusions

Previous research has consistently shown the efficacy of utilizing the CRISPR/Cas9 system and NHEJ repair mechanism to induce mutations in rice plants, resulting in improved resistance to abiotic stress factors.

This study was conducted to confirm the biological and genetic functions of the U-box type E3 ubiquitin ligase gene against biotic stress in rice. CRISPR/Cas9 technology was used to make a gene editing lines that lost the function of the PUB gene, and association of stress resistance and function of the *OsPUB9* gene were confirmed through infection treatment of *Xoo* strains that cause leaf bacterial leaf blight. *OsPUB9* gene-specific sgRNA were designed and transformants were developed through *Agrobacterium*-mediated transformation. Deep sequencing using callus was performed to confirm the mutation type of T_0_ plants, and a total of three steps were performed to select null individuals without T-DNA insertion. In the case of the *OsPUB9* gene-edited line, a one bp insertion was generated by gene editing, and it was confirmed that early stop codon and multiple ORF sites were created by inserting thymine. It is presumed that the ubiquitination function also changed according to the change in protein structure of the U-box E3 ubiquitin ligase. The *OsPUB9* gene-edited null lines were inoculated with bacterial leaf blight, and finally confirmed to have a resistance phenotype similar to Jinbaek rice, a rice blight-resistant cultivar. Therefore, it is assumed that the amino acid sequence derived from the *OsPUB9* gene is greatly changed, resulting in a loss of protein functions related to biological mechanisms. Comprehensively, it was confirmed that resistance to bacterial leaf blight stress is enhanced when a mutation occurs at a specific site in the*OsPUB9* gene.

The ubiquitination process has emerged as a promising target for enhancing crop stress tolerance [65,66]. The abundance, specificity, and diverse structural characteristics of various E3-ubiquitin ligases suggest their versatility and potential involvement in numerous cellular processes [67]. Future research should be able to focus on further exploring the complex mechanisms regulating the genes and proteins responsible for a wide range of stress tolerance under this premise.

## Figures and Tables

**Figure 1 ijms-25-07145-f001:**
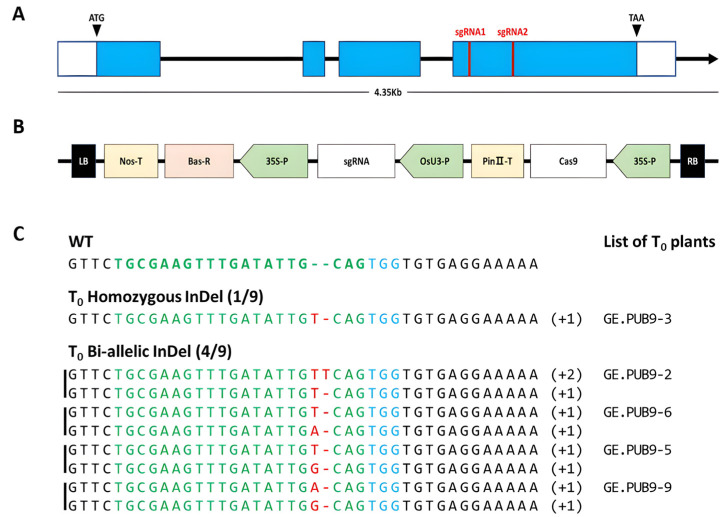
Detection of *OsPUB9* gene-edited lines. (**A**) Structure of the *OsPUB9* gene and selection of the target region of *OsPUB9* for the CRISPR RGEN tool. Blue box indicates the exon region, and white box indicates the 5′-UTR and 3′-UTR, respectively. (**B**) Construction of the Ti-plasmid vector of the sgRNA region for CRISPR/Cas9-mediated mutagenesis of *OsPUB9* genes in rice. LB, left border; RB, right border; Nos-T, nopaline synthase terminator; Bas-R, basta resistance gene; sgRNA, single guide RNA of *OsPUB9*; *OsU3*-P, *Oryza sativa* U3 promoter driven sgRNA cassette; and Cas9, human codon-optimized Cas9 expressing cassette. (**C**) Identification of mutation genotypes in the *OsPUB9* gene generated by CRISPR/Cas9 technology through NGS analysis in WT and T_0_ plants. Green indicates the target sequence, and blue indicates the PAM region in WT. InDel sequence of the target locus is shown in red. Insertion is indicated by an insertion of nucleotides such as “T” or “G”. The plus (+) signs indicate the number of nucleotides inserted at the target sites, respectively. WT, wild type sequence with no mutations.

**Figure 2 ijms-25-07145-f002:**
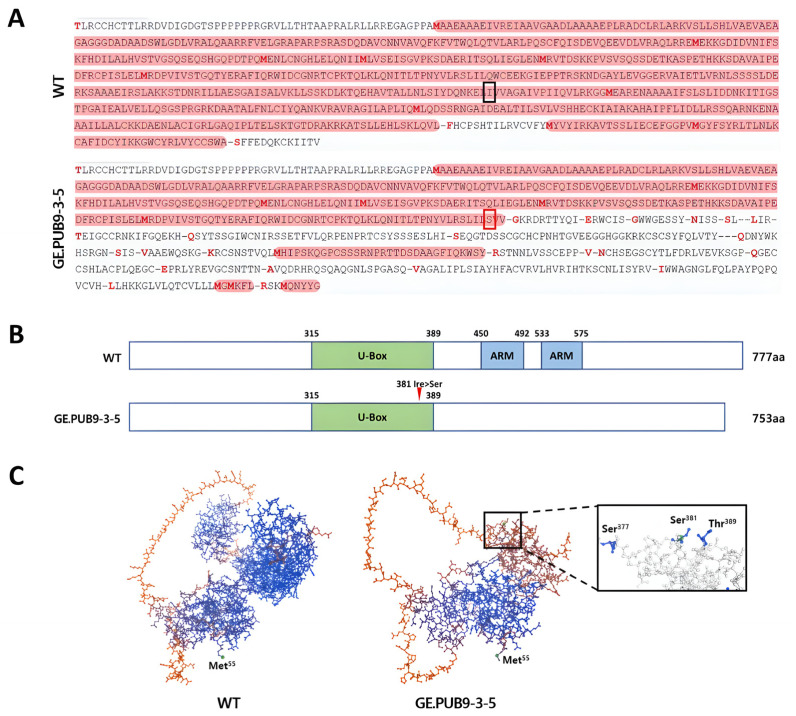
(**A**) Predicted amino acid sequence based on nucleotide sequence of *OsPUB9* gene-edited null line with *OsPUB9* wild type. The red border box indicates the region where one bp insertion occurred due to gene editing of *OsPUB9* gene, and the red highlight indicates ORFs that frequently occurred due to nucleotide mutation. (**B**) Schematic diagram of the *OsPUB9* gene, showing domains and amino acid changes induced by CRISPR/Cas9. U-box, U-box domain; ARM, Armadillo (ARM) repeat domain. (**C**) Putative protein structure based on nucleotide sequence of *OsPUB9* gene-edited null line (GE.PUB9-3-5) with *OsPUB9* (Os02g0732200).

**Figure 3 ijms-25-07145-f003:**
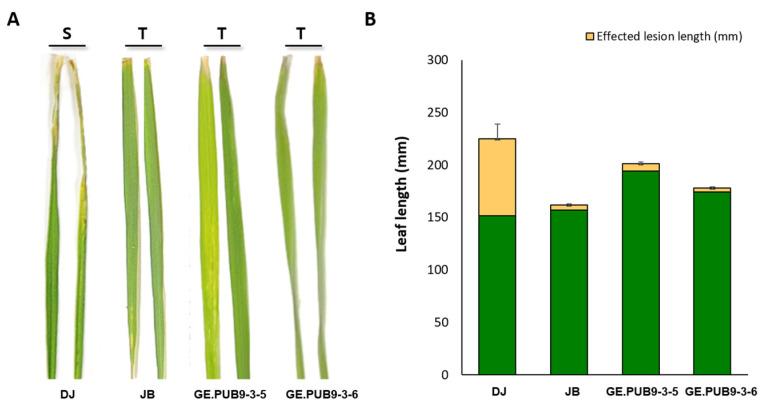
Phenotype and lesion length of bacteria blight (race K2) inoculation in Dongjin (DJ) and Jinbaek (JB), T_2_ *OsPUB9* gene-edited null lines. (**A**) Phenotypic reaction of plants. (**B**) Average lesion length of leaf bacterial leaf blight symptoms measured in mm taken at 14 days post inoculation.

**Figure 4 ijms-25-07145-f004:**
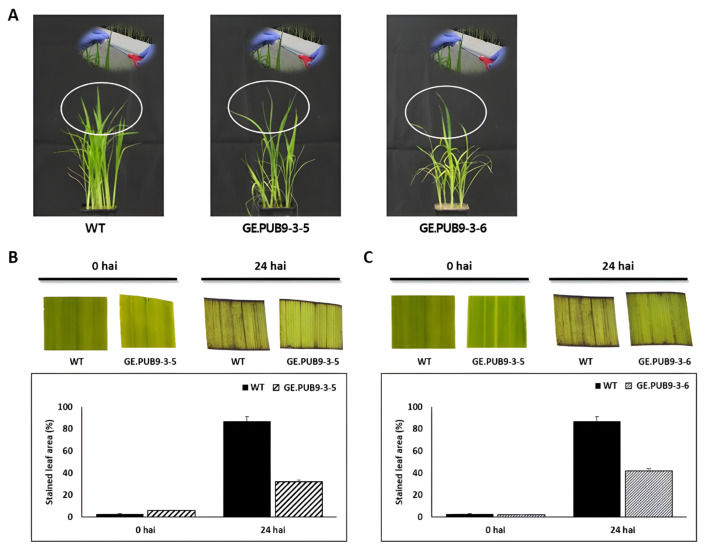
H_2_O_2_ detection with DAB staining in leaf segments of WT and *OsPUB9* gene-edited null lines (GE.PUB9-3-5 and GE.PUB9-3-6) in response to bacterial blight stress. (**A**) Phenotype of flag leaves during ripening phase used in H_2_O_2_ detection with DAB staining. (**B**) Histochemical detection of H_2_O_2_ after 24 h post-inoculation and quantitative accumulation of leaf areas stained in GE.PUB9-3-5. (**C**) Histochemical detection of H_2_O_2_ after 24 h post-inoculation and quantitative accumulation of leaf areas stained in GE.PUB9-3-6. Three replicates were used, and data are shown with SE. hai, hours after inoculation.

**Figure 5 ijms-25-07145-f005:**
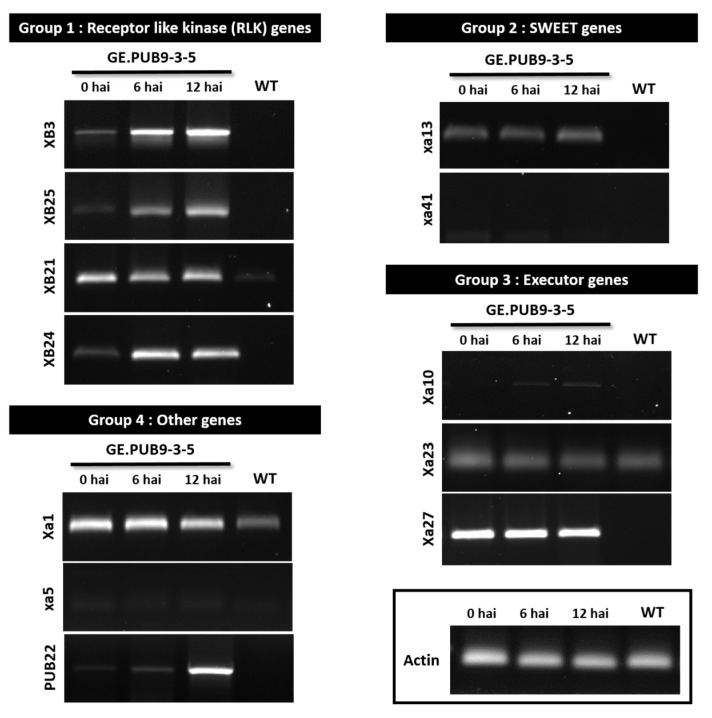
Expression analysis of the cloned rice R genes and the cognate *Xanthomonas oryzae Avr* genes according to biotic stress treatment in the T_2_ GE.PUB9-3-5 gene-edited line. Confirmation of mRNA expression level with bacterial leaf blight resistance-related genes using the T_2_ *OsPUB9* gene-edited line 0, 6, 12 h after K2 strain inoculation (HAI) in agarose gel electrophoresis. hai, hours after inoculation.

**Figure 6 ijms-25-07145-f006:**
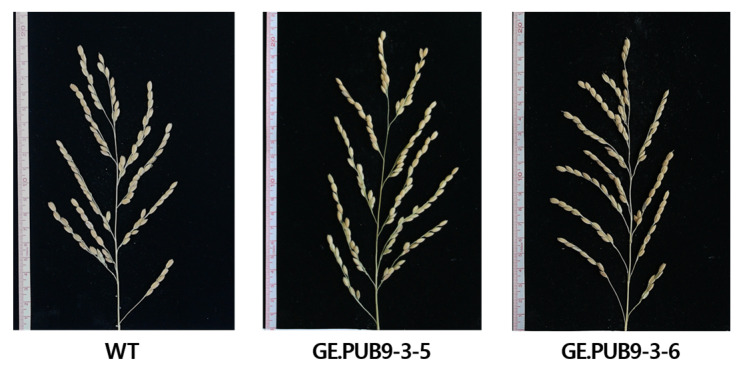
Phenotypes of panicle-selected *OsPUB9* gene-edited null lines in the T_3_ generation.

**Table 1 ijms-25-07145-t001:** Selection of sgRNA of the *OsPUB9* gene using CRISPR RGEN tools.

Target gene	sgRNA	RGEN Target (5′ to 3′)	Direction	Cleavage * Position (%)	GC Contents (%, w/o PAM)	Out-of-Frame Score	Mismatches (bp)		
							0	1	2
*OsPUB9*	sgRNA1	TGCGAAGTTTGATATTGCAGTGG	+	50.2	40	88.8	1	0	0
	sgRNA2	TTTGATATTGCAGTGGTGTGAGG	+	53.1	40	79.9	1	0	0

* Cleavage position (relative position of cleavage site of the selected RGEN) in the target searching ranges. If multiple ranges are specified, then the non-specified ranges are neglected.

**Table 2 ijms-25-07145-t002:** Agronomic traits of *OsPUB9* gene-edited null lines in the T_3_ generation.

Rice Line	Plant Height (cm)	Culm Length (cm)	Panicle Length (cm)	No. of Tiller
Dongjin (WT)	114.0 ± 1.26	94.3 ± 1.60	19.3 ± 0.90	12 ± 0.50
GE.PUB9-3-5 ^ns^	115.7 ± 1.03	91.9 ± 2.71	19.8 ± 0.25	11 ± 0.81
GE.PUB9-3-6 ^ns^	123.5 ± 1.47	91.8 ± 2.37	19.0 ± 0.76	11 ± 0.82

ns, not significant.

## Data Availability

Data are contained within the article.

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
