# Peer review of "OsPUB9 Gene Edited by CRISPR/Cas9 Enhanced Resistance to Bacterial Leaf Blight in Rice (Oryza sativa L.)"

_ijms, 2024, doi:10.3390/ijms25137145_

Round 1

Reviewer 1 Report

Comments and Suggestions for Authors

The authors performed gene edition of OsPUB9 by CRISPR/Cas9. One transgenic line was found to have enhanced resistance to bacterial leaf blight in rice. The manuscript was well- written. However, if the bacterial blight resistance was induced by the functional inactivation of PUB9, multiple transgenic lines should have similar resistance phenotype. Current results only displayed bacterial blight resistance for PUB9-3, which could be induced by variations in other genes on the genome. How about the bacterial blight resistance for PUB9-2, PUB9-6, PUB9-5, PUB9-9? The authors should solve this problem.

Other problems were listed below:

In 2.1.1., ‘T0 9 transformants for OsPUB9 gene’ should be ‘9 T0 transformants for OsPUB9 gene’.

In 2.1.3, whether the amino acid sequence of OsPUB9 would be change for PUB9-2, PUB9-6, PUB9-5, PUB9-9?

In 2.2., whether bacterial blight inoculation were conducted for PUB9-2, PUB9-6, PUB9-5, PUB9-9? If it was done, show the result.

In 2.3., ‘Of the 11 R genes nine R genes’ should be ‘Of the 11 R genes, nine R genes’

In 2.4. , for agronomic traits, how about the changes of number of filled grain per plant, number of total grains per plant, seed setting rate, 1000 grain weight

Comments on the Quality of English Language

The English is writtern well.

Author Response

Response to Reviewer 1

The authors performed gene edition of OsPUB9 by CRISPR/Cas9. One transgenic line was found to have enhanced resistance to bacterial leaf blight in rice. The manuscript was well- written. However, if the bacterial blight resistance was induced by the functional inactivation of PUB9, multiple transgenic lines should have similar resistance phenotype. Current results only displayed bacterial blight resistance for PUB9-3, which could be induced by variations in other genes on the genome. How about the bacterial blight resistance for PUB9-2, PUB9-6, PUB9-5, PUB9-9? The authors should solve this problem.

-- Thank you for your kind comments. In response to the reviewer's in-depth question, we present the results of Xoo inoculation with PUB9-6 (A-) and PUB9-5 (G-) derived from T2 plants in the supplementary figure 5, which was similar to the results of GE-PUB9-3-5.

Other problems were listed below:

In 2.1.1., ‘T0 9 transformants for OsPUB9 gene’ should be ‘9 T0 transformants for OsPUB9 gene’.

-- Thank you for your kind comments. We revised it in lines 150-152 as follows: ‘CRISPR/Cas9-OsPUB9 vector was inoculated into callus obtained from Dongjin (wild-type) seeds using Agrobacterium-mediated method to generate a total of 9 T0 transformants for OsPUB9 gene.

In 2.1.3, whether the amino acid sequence of OsPUB9 would be change for PUB9-2, PUB9-6, PUB9-5, PUB9-9?

--Thank you for your kind comments. We presented the sequence data of amino acids of PUB9-2, PUB9-6, PUB9-5, and PUB9-9 in Supplementary Figure 4.

In 2.2., whether bacterial blight inoculation was conducted for PUB9-2, PUB9-6, PUB9-5, PUB9-9? If it was done, show the result.

--Thank you for the reviewer's point. We presented the results of Xoo inoculation with GE.PUB9-2, GE.PUB9-5, GE.PUB9-6, GE.PUB9-9 in the supplementary figure 5.

In 2.3., ‘Of the 11 R genes nine R genes’ should be ‘Of the 11 R genes, nine R genes’

--Thank you, we have revised it according to the reviewer pointed out.

In 2.4. for agronomic traits, how about the changes of number of filled grain per plant, number of total grains per plant, seed setting rate, 1000 grain weight

--Thank you for your kind comments. The agronomic traits of OsPUB9 gene-edited null lines in the T3 generation were similar to those of the wild type Dongjin.

Reviewer 2 Report

Comments and Suggestions for Authors

Manuscript titled “OsPUB9 Gene Edited by CRISPR/Cas9 Enhanced Resistance to 2 Bacterial Leaf Blight in Rice (Oryza sativa L.)” investigated the role of OsPUB9 in rice against bacterial leaf blight through disrupting or modifying OsPUB9 gene function through genome editing. The authors generated null transformants in rice and evaluated their resistance against bacterial leaf blight. However, there are some concerns that the authors should address in order to support their investigations. It looks authors were not  attentive while writing the manuscript, it is crucial to maintain consistency between the results and experimental methodologies presented, in the manuscripts there are contradictions to your methodologies and results presented.  

Major comment

Introduction

The authors should consider rewriting and organising the introduction to focus specifically on the research topic. The introduction should highlight the role of the ubiquitin-ligase system in plant defence mechanisms with relevant recent studies that demonstrate the ubiquitin-ligase system's importance in enhancing plant resistance against biotic stresses. This will help the broad-spectrum readers to understand the importance of the study.

Result

2.1.1. Generation of knockout mutant lines of OsPUB9 gene

When the nucleotide sequence  of Os02g0732200 was searched against a rice database using BLASTN, it showed a 100 % match with  LOC_Os02g49950 in the rice genome project. Are these two genes are same? If so, can you provide the chromosome location of these genes? If they are homologs, have you considered this while designing the guide RNAs?  

2.1.2. Analysis of mutation types through deep-sequencing and generation of OsPUB9 146 gene-edited null lines

Did the authors perform null-line analysis for the 9 line or the selected best line? – specify this in the results

Is the 3:1 ratio derived from the single line?

2.3. Expression analysis of R genes related to biotic stress resistance

There are no sufficient controls to validate the results in this section effectively. The authors should collect the samples at the same time points from both transgenic lines without infection and wild-type with and without infection. By considering these controls, authors can make roust comparison and provide stronger support to their hypothesis.so it is crucial to revise this results section to include the necessary controls

The authors mentioned using qRT-PCR in the materials and method section. However, in the results, they presented agarose gel pictures showing gene expressions. It is important to clarify the reason for this discrepancy and ensure that methods and results should align properly.

In the result section, the authors should only focus on presenting  their experimental results rather than discussing  the functions of the genes.             

2.4. Investigation of agronomic traits

Figure 6: why is the seed coat colour of the GE.PUB9-3-5 brighter than that of the other two plants ?

4.7. Analysis of DAB staining

Why was the third leaf of seedling selected?  Is it inoculated with bacterial culture? There would be many tillers in which tiller 3rd leaf was used.  

Did you use the entire leaf for the DAB staining?

Did the authors really used seedlings, if so, why did the authors representing a mature plant (at the ripening stage) in Figure 4A .

It is important to clarify the reason for this discrepancy and ensure that methods and results should align properly.

 4.8. Expression analysis of R genes related to biotic stress resistance

Describe in detail the process of sample collection from transgenic and wild-type plants. In addition, write the methodology used for  RNA extraction and cDNA synthesis.

Minor comments

Authors have to maintain the scientific writing throughout the MS, in many cases gene names are not italicized, please check carefully and bring the changes   

Line 142: Supplementary Figure 1: authors may have considered wild-type plant DNA for the negative control   

Line 163: Fig 1A- describe the color scheme

Line 224-225: rewrite the sentence as it is currently unclear

Line 410: Agrobacterium- ensure uniformity throughout the MS  

Line 433: T0 select null lines without T-DNA, 14 days old seedlings.. should be written as ”Tto select T0 null lines……”

Line 448-450: it is currently unclear, rewrite it clearly what the authors want to say.   

Discussion

Avoid repeating results instead focus on discussing probable reasons for the results and supporting them with previous studies.    

Comments on the Quality of English Language

in the manuscript, in several places, the sentences were not clear. I have mentioned some of the examples in my comment. I suggest authors to consider editing the manuscripts with a native English speaker.  

Author Response

Response to Reviewer 2

Manuscript titled “OsPUB9 Gene Edited by CRISPR/Cas9 Enhanced Resistance to 2 Bacterial Leaf Blight in Rice (Oryza sativa L.)” investigated the role of OsPUB9 in rice against bacterial leaf blight through disrupting or modifying OsPUB9 gene function through genome editing. The authors generated null transformants in rice and evaluated their resistance against bacterial leaf blight. However, there are some concerns that the authors should address in order to support their investigations. It looks authors were not attentive while writing the manuscript, it is crucial to maintain consistency between the results and experimental methodologies presented, in the manuscripts there are contradictions to your methodologies and results presented. 

Major comment

Introduction

The authors should consider rewriting and organising the introduction to focus specifically on the research topic. The introduction should highlight the role of the ubiquitin-ligase system in plant defence mechanisms with relevant recent studies that demonstrate the ubiquitin-ligase system's importance in enhancing plant resistance against biotic stresses. This will help the broad-spectrum readers to understand the importance of the study.

-- Thank you for your comments. We modified sentences in line 43 through 58 for introduction.

Result

2.1.1. Generation of knockout mutant lines of OsPUB9 gene

When the nucleotide sequence of Os02g0732200 was searched against a rice database using BLASTN, it showed a 100 % match with LOC_Os02g49950 in the rice genome project. Are these two genes are same? If so, can you provide the chromosome location of these genes? If they are homologs, have you considered this while designing the guide RNAs? 

-- Thank you for your comments. As the reviewer pointed out, it's just the same gene with different IDs. That is, Os02g0732200 is the RAP ID, and LOC_Os02g49950 is the MSU ID.

2.1.2. Analysis of mutation types through deep-sequencing and generation of OsPUB9 gene-edited null lines

Did the authors perform null-line analysis for the 9 line or the selected best line? – specify this in the results

Is the 3:1 ratio derived from the single line?

-- Thank you for your comments. For the gene-edited individuals we obtained, the edited individuals were selected as homo- and bi-allelic in the T0 generation, and T1 seeds were harvested from these lines. T2 seeds were cultivated in the field by selecting susceptible individuals through basta treatment on the leaves of plants obtained by sowing more than 30 T1 seeds. Of course, the ratio of the resistant and susceptible individuals of the basta was separated by 3:1 (Chi Square test result).

2.3. Expression analysis of R genes related to biotic stress resistance

There are no sufficient controls to validate the results in this section effectively. The authors should collect the samples at the same time points from both transgenic lines without infection and wild-type with and without infection. By considering these controls, authors can make roust comparison and provide stronger support to their hypothesis. so it is crucial to revise this results section to include the necessary controls

-- Thank you for your comments. We analyzed the expression level of R genes over time using OsPUB9 gene editing line, and as a result, we confirmed that most R genes showed strong expression in 12-hour after inoculation. Based on these results, the analysis was conducted based on the 12-hour post-inoculation WT sample as a control in which most R genes were strongly expressed.

The authors mentioned using qRT-PCR in the materials and method section. However, in the results, they presented agarose gel pictures showing gene expressions. It is important to clarify the reason for this discrepancy and ensure that methods and results should align properly.

-- Thank you for your kind comments. In Materials and methods, qRT-PCR was modified to RT-PCR.

In the result section, the authors should only focus on presenting their experimental results rather than discussing the functions of the genes.

-- Thank you for the reviewer's advice. We analyzed the expression of R genes in OsPUB9 gene editing line. According to previous research, each R gene has a different role in disease resistance in terms of genetic structure, so this study also sufficiently shows that it is related to the high disease resistance in the OsPUB9 gene-edited lines.

2.4. Investigation of agronomic traits

Figure 6: why is the seed coat colour of the GE.PUB9-3-5 brighter than that of the other two plants?

-- Thank you for your kind comments. Maybe it looked like that because the degree of exposure of light was different, so we replaced the picture.

4.7. Analysis of DAB staining

Why was the third leaf of seedling selected?  Is it inoculated with bacterial culture? There would be many tillers in which tiller 3rd leaf was used. 

Did you use the entire leaf for the DAB staining?

Did the authors really used seedlings, if so, why did the authors representing a mature plant (at the ripening stage) in Figure 4A.

It is important to clarify the reason for this discrepancy and ensure that methods and results should align properly.

-- Thank you for your kind comments. We modified sentences line 482 to 491 in ‘4.7. Analysis of DAB staining’.

4.8. Expression analysis of R genes related to biotic stress resistance

Describe in detail the process of sample collection from transgenic and wild-type plants. In addition, write the methodology used for RNA extraction and cDNA synthesis.

-- Thank you for your kind comments. We modified sentences line 494 to 521 in ‘4.8. Expression analysis of R genes related to biotic stress resistance’.

Minor comments

Authors have to maintain the scientific writing throughout the MS, in many cases gene names are not italicized, please check carefully and bring the changes  

Line 142: Supplementary Figure 1: authors may have considered wild-type plant DNA for the negative control  

Line 163: Fig 1A- describe the color scheme

Line 224-225: rewrite the sentence as it is currently unclear

Line 410: Agrobacterium- ensure uniformity throughout the MS 

Line 433: T0 select null lines without T-DNA, 14 days old seedlings.. should be written as ”Tto select T0 null lines……”

Line 448-450: it is currently unclear, rewrite it clearly what the authors want to say.

-- Thank you for your kind comments. We modified each minor parts. 

Discussion

Avoid repeating results instead focus on discussing probable reasons for the results and supporting them with previous studies.  

-- Thank you for your kind comments. We deleted the explanation of repeated experimental results and further discussed previous related studies and the results of this study.

Round 2

Reviewer 1 Report

Comments and Suggestions for Authors

The author answered all my concerns.

Reviewer 2 Report

Comments and Suggestions for Authors

The article is recommended for publication.